# Moonless night sky increases *Isistius* species (cookiecutter shark) and live human contact

**Steven Minaglia**[1]*, **Melodee Liegl**[2]

1 Division of Urogynecology, Department of Obstetrics & Gynecology, Queen's University Medical Group and John A. Burns School of Medicine, University of Hawaii, Honolulu, Hawaii, United States of America,
2 Departments of Pediatrics, Quantitative Health Sciences, Medical College of Wisconsin, Milwaukee, Wisconsin, United States of America

☯ These authors contributed equally to this work.
* minaglia@hawaii.edu

**Data Availability Statement:** All relevant data are within the manuscript and its Supporting Information files.

**Funding:** The author(s) received no specific funding for this work.

## Abstract

The nocturnal feeding behavior and zoogeographical habitat of cookiecutter sharks *Isistius brasiliensis* and *Isistius plutodus* (*Isistius spp.*) greatly reduces interaction of this species with live humans. Attacks on live humans are exceedingly rare with 7 cases reported worldwide, 6 of them in Hawai'i, and 5 of these occuring among channel swimmers. Published research suggests that periods of bright moonlight may increase *Isistius spp.* contact with live humans and does not otherwise identify significant trends or risk factors. Yet 5 of the 6 *Isistius spp.* bites on live humans in Hawai'ian waters occurred with the moon set and after nautical twilight end and before nautical twilight start. From 1961–2023 in Hawai'i, 129 successful solo channel crosses and 5 *Isistius spp.* related injuries in the habitat of cookiecutter sharks were analyzed across two groups: one where both the moon and sun were set (dark group) and one where the moon and/or sun was in the sky (light group). There was a significant difference for swimmers bitten by *Isistius spp.* in the dark 4 (12%) versus light groups 1 (1%), p = 0.012, RR 12.6 (95% confidence interval: 1.5–108.9). Swim start time and year was also significant (Pearson correlation 0.566, p <0.001). Swimmer gender and use of shark deterrent devices and artificial illumination were not significant. The growing popularity of channel swimming in Hawai'i and swim start times have contributed to an increasing likelihood of live human and *Isistius spp.* contact and a moonless night sky is a significant risk factor for this interaction.

## Introduction

The two species of cookiecutter sharks, *Isistius brasiliensis* and *Isistius plutodus* (*Isistius spp.*) are best known for the characteristic wound they leave on their prey items and for their "hit and run" feeding behavior recently witnessed by live humans [1, 2]. These pelagic sharks remove circular plugs of flesh by sucking onto their prey using specialized pharyngeal muscles and fleshy lips, inserting their upper jaw teeth, and sweeping their proportionately larger lower jaw teeth through the flesh often twisting in a circular motion to remove and ingest their feed [3]. The resultant circular, concave wound is a characteristic sign of an *Isistius spp.* bite.

**Competing interests:** The authors have declared that no competing interests exist.

Cookiecutter sharks are thought to undertake diel vertical migrations with a daytime depth range between 1,000–3,500 meters in primarily warm, coastal waters near islands [4]. They typically rise near the surface at night to feed on squid, small fish, and a variety of large pelagic species and then return to the bathyal zone when daylight resumes [5–7]. They possess a well-developed lateral line system, bioluminescence, and large eyes that give them a competitive advantage over their prey especially in low-light settings.

Contact between *Isistius spp.* and deceased humans afloat in the sea has been documented [8–11] yet contact with live humans is exceedingly rare. Out of 7 ever-documented and confirmed attacks on live humans, 6 occurred in Hawai'ian waters between the years 2009 and 2023 and 5 of these occurred among channel-swimmers [1, 2, 12, 13]. Channels are bodies of water that separate land forms. Channel swimming is the sport of swimming across these large bodies of water from one land form to another. Hawai'i is home to 9 channels with varying distances and depths. Five of these channels match the zoogeographical habitat of *Isistius spp.* and have been traversed by channel swimmers (Table 1). All 6 *Isistius spp.* bites in Hawai'i have occurred in two of these channels, the 'Alenuihāhā Channel being the 2nd deepest and the Kaiwi Channel being the 4th deepest in Hawai'i (Table 2). The minimum surface over water depth where one of the 6 bites in Hawai'i occurred was estimated at 366 meters [13].

Published research suggests that periods of bright moonlight may increase *Isistius spp.* contact with live humans [1, 14]. A 2022 National Geographic documentary film focused on a spike in attacks in 2019 but did not otherwise identify significant trends or risk factors [15]. Yet 5 out of 6 documented *Isistius spp.* bites on live humans in Hawai'ian waters occurred with the moon set and after nautical twilight end and before nautical twilight start. The research question therefore generated from this observation is whether absence of the sun and moon from the night sky while a human swimmer transits through the zoogeographical zone of *Isistius spp.* is an independent risk factor for contact with the species and potential injury.

## Materials and methods

A retrospective review of the State of Hawai'i Department of Land and Natural Resources (DLNR) Division of Aquatic Resources (DAR) Shark Incidents List [13] identified 6 live

**Table 1. Characteristics of Hawaii's 5 deepest channels.**

| Channel | Maximum Depth (meters) | Successful Crosses | Victims |
|---|---|---|---|
| 'Alenuihāhā | 2149 | 4 | 1 |
| Ka'ie'iewaho | 3319 | 1 (relay) | 0 |
| Kaulakahi | 1273 | 19 | 0 |
| Kaiwi | 680 | 101 (5 relays) | 4 |
| Kealaikahiki | 390 | 5 | 0 |

**Table 2. Surface over water depth during attacks.**

| Victim Number | Surface over water depth (meters) |
|---|---|
| 1 | >610 |
| 2 | >610 |
| 3 | >610 |
| 4 | >610 |
| 5* | 366 |
| 6 | 549 |

*non-channel swimmer

victims of *Isistius spp.* one of whom was not swimming a channel. A comprehensive Web of Science and Pubmed search of the literature written in English using the terms: "*Isistius* species", "*Isistius sp.*", "*Isistius spp.*", "cookiecutter shark", "cookiecutter shark bite", "shark bite", and "Hawai'i" on July 1, 2023 was also performed and confirmed the above 6 victims to be the only documented and confirmed cases of *Isistius spp.* attacks on live humans in Hawai'i. The International Shark Attack File identified these victims as well as another live victim in North Queensland, Australia in 2017 [12].

This retrospective cohort study included all channel swimmers that were either injured by *Isistius spp.* and/or successfully completed a solo crossing of any of the 5 Hawai'ian channels that match the zoogeographical natural habitat of *Isistius spp.* With the exception of one swimmer, data for all swimmers including victims of attacks was obtained from publicly available, previously published sources [1, 2, 12, 13, 15, 16]. Specifically, the date of swim, duration of swim, gender, start and end times, and use of artificial lights and/or electronic shark deterrents were known for successful swimmers and are part of an established database used to certify and celebrate the accomplishment of the swimmers [16]. Additionally, data for 4 of the victims was complete and obtained from the case report and case series that described their attacks [1, 2]. Written informed consent was obtained from the sole successful yet injured swimmer described in this report (Fig 3., Table 2: Victim 6) whom provided additional data that was not publicly available at the time this study was performed. The Institutional Review Board policy of the Queen's Medical Center does not require approval for use of the information pertaining to the one swimmer. All successful solo crossings were included in the analysis even if they were tandem solo crossings. One successful relay swim across the Ka'ie'iewaho Channel and 5 successful relay swims across the Kaiwi channel were excluded from analysis due to multiple swimmers swimming in series with varying speeds, varying use of artificial illumination, and varying use of shark deterrent devices. Swims across the 'Alenuihāhā, Kaulakahi, Kaiwi, and Kealaikahiki channels were thus included in the analysis.

The minimum surface over water depth where one of the 6 recorded *Isistius spp.* bites on live humans had occurred in Hawai'i was used as the benchmark to determine where the zoogeographical zone of *Isistius spp.* existed over the swimming course for each of Hawai'i's 4 deepest and crossed-by-solo-swimmer channels [13]. Water depth was estimated from National Oceanic and Atmospheric Administration (NOAA) Office of Coast Survey charts 19010 and 19380 [17, 18]. Additionally, the exact location of each of the 5 channel-swimming victims at which time their injury occurred is known and lies within this calculated zoogeographical zone [1, 2, 12, 13]. The line distance of each swim course for each channel and the total swim time was known for all 129 completed swims occurring in the 4 channels from September 1961- July 1, 2023. Speed in statute miles per hour was calculated for the entirety of each completed swim and the estimated time each swimmer entered and exited the calculated zoogeographical zone of *Isistius spp.* was recorded.

Additional data included nautical twilight start and end times and moon rise and set times [19–21]. Swimmers transiting for any length of time through the above determined zoogeographical zone of *Isistius spp.* after nautical twilight end and with the moon set and before nautical twilight start were grouped as the 'dark group'. The remaining swimmers comprised the 'light group' because they transited through this zone with the moon up and/or after nautical twilight start or before nautical twilight end.

Non-parametric Fisher's exact test was performed for categorical variables (reported as N (%)). Pearson correlation coefficient was calculated to examine the relationships between start time of swim and year of channel swim. A p-value $<0.05$ was considered significant. SPSS version 28 (Chicago, Illinois, USA) was used for analyses.

## Results

Review of the State of Hawaii Department of Land and Natural Resources (DLNR)- Division of Aquatic Resources (DAR) Shark Incidents List identified 6 live victims of *Isistius brasiliensis* and *Isistius plutodus* (*Isistius spp.*) one of whom was not swimming a channel. The individual not swimming a channel was attacked in the ʻAlenuihāhā Channel at an estimated surface over water depth of 366 meters and similar to 4 out of 5 of the channel-swimming victims was attacked with the moon set and after nautical twilight end and before nautical twilight start. Given the individual has remained confidential and was not participating in a channel-swim he/she has been excluded from analysis. Overall, the date of swim, duration of swim, start and end times, gender, use of artificial lights and/or electronic shark deterrents, nautical twilight start and end times and moon rise and set times were obtained for all 133 swimmers either successful and/or injured.

Of the 133 swimmers the majority were male (65%). The most common channel swim was Kaiwi (78%), followed by Kaulakahi (14%), Kealaikahiki (4%) and ʻAlenuihāhā (4%). All but 2 swimmers swam from Molokaʻi to Oʻahu to complete the Kaiwi Channel, all but 3 swimmers swam from Kauaʻi to Niʻihau to complete the Kaulakahi Channel, all swimmers swam from Hawaiʻi to Maui to complete the ʻAlenuihāhā channel and all swimmers swam from Kahoʻolawe to Lānaʻi to complete the Kealakahiki channel. 123 swimmers and/or crew (92%) were successfully contacted to obtain the additional data pertaining to use of artificial illumination and/or electronic shark deterrent devices. Overall, 111 (83%) of swimmers and/or crew were able to recall the use of artificial illumination and/or the use of electronic shark deterrent devices during their swims. 73% of the swimmers used artificial illumination. Electronic shark deterrent devices were used for over half the swimmers (53%). The sun was up for 56% of the swimmers and the sky was lit by moon or sun for the majority (76%).

A total of 5 swimmers (4%) encountered 7 *Isistius spp.* bites (Figs 1 and 2). 3 of the bitten swimmers used electronic shark deterrent devices. 3 of the bitten swimmers used glow sticks while the other 2 did not use any form of artificial illumination. Of note, boat lights went on shortly before the latter 2 swimmers were attacked. One swimmer was injured by two distinct *Isistius spp.* bites and was still able to complete the Kaiwi Channel (Fig 3). There was a significant difference for the swimmers bitten by *Isistius spp.* in the dark group 4 (12%) versus the light group 1 (1%), p = 0.012 (Table 3). The relative risk was 12.6 (95% confidence interval) 1.5–108.9). There were no significant differences when comparing dark and light sky with gender, use of shark deterrent devices and the presence of artificial illumination (sundown only). There were no significant differences when comparing those receiving no bite versus bite with gender, electronic shark deterrent devices, artificial illumination (sundown only). All 5 (100%) bites occurred with sun down versus the 70 (55%) swimmers in the no bite group. This was marginally significant at p = 0.068 (Table 4).

The two channels, Kaulakahi and Kealaikahiki are both 17 miles and on average took swimmers 11 hours 17 minutes (range 9:25–14:28) and 11 hours 22 minutes (range 10:24–11:53) to cross, respectively. The ʻAlenuihāhā and Kaiwi Channels are 30 and 26 miles in length and have average crossing times of 17 hours 26 minutes (range 14:51–20:08) and 16 hours 40 minutes (12:02–27:33), respectively. Swim start time and year of channel swim was significant when looking at the ʻAlenuihāhā and Kaiwi channels only. During the earlier years, the start time was just after midnight versus later years (post 2010) starting closer to 6 PM. Pearson correlation was 0.566, p <0.001 (Fig 2). There was no significant correlation between moon phase in days and year of swim for ʻAlenuihāhā channel (p = 0.723) and Kaiwi channel (p = 0.140) and both combined (p = 0.225)(Fig 5). In contrast, all swim start times were between 6:00 AM and 7:33 AM for the Kealaikahiki Channel and between 4:21 AM and 6:27 AM for the

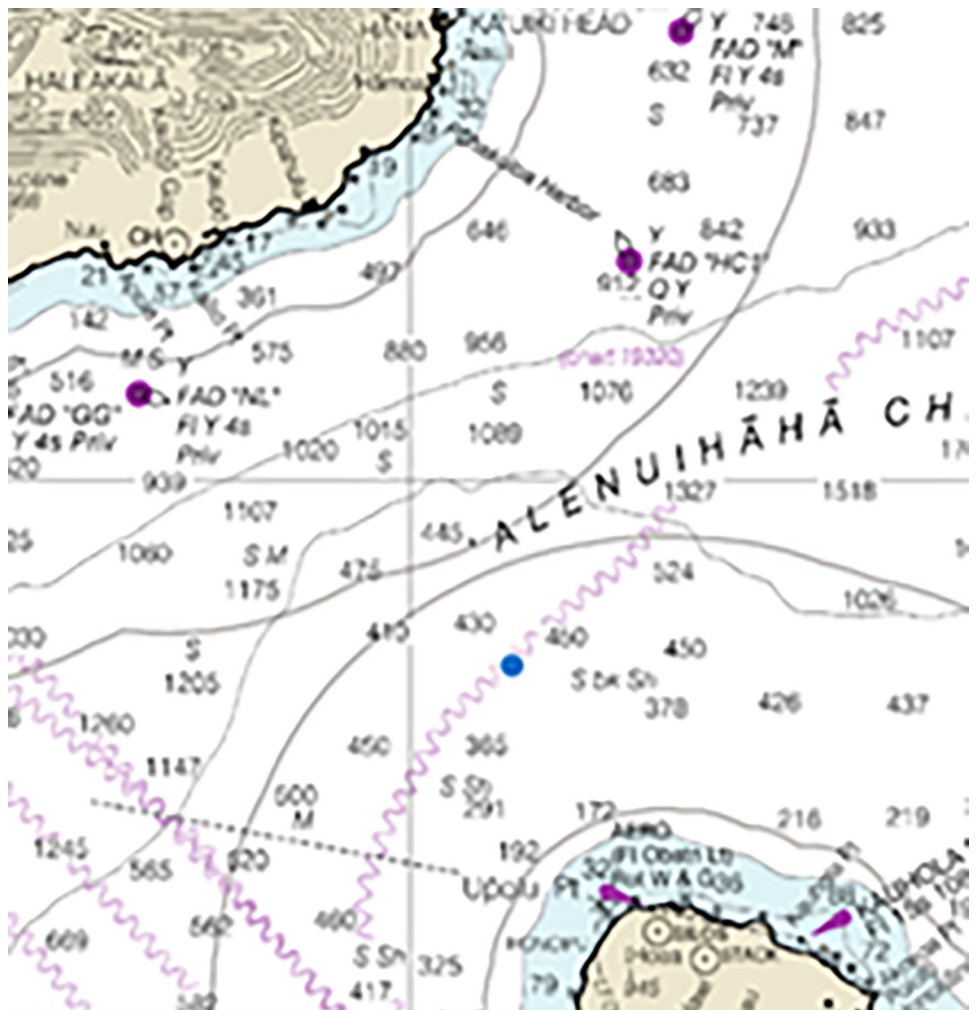

**Fig 1. Map with depth isobaths in fathoms showing the location of the attack in ʻAlenuihāhā Channel (Blue dot victim 1).** Provided by National Oceanic and Atmospheric Administration (NOAA) Office of Coast Survey, nauticalcharts.noaa.gov.

Kaulakahi Channel. This effectively placed all swimmers of these two channels in the zoogeographical habitat of Isistius spp. during daytime.

## Discussion

The width of each channel determines the average time it takes for a channel swimmer to cross. The Kaiʻeiʻewaho Channel is 72 miles and has only been crossed once by a relay team of 6 swimmers with a total time of 47 hours 55 minutes. This channel is the deepest and given its length it is unlikely to be attempted by a serious solo swimmer. In 2008, the inclusion of the Kaiwi Channel to Oceans 7, a list of 7 long-distance open-water swims through some of the worlds' most dangerous sea channels has led to a significant increase in swims across this particular channel. It is highly unlikely for any swimmer to complete either the ʻAlenuihāhā or Kaiwi channels exclusively during daylight hours given average crossing times.

*Isistius spp.* migrates to surface waters at night and during the day to depths up to 3500 meters. It feeds on large pelagic fishes with its unique "hit and run" feeding behavior and also

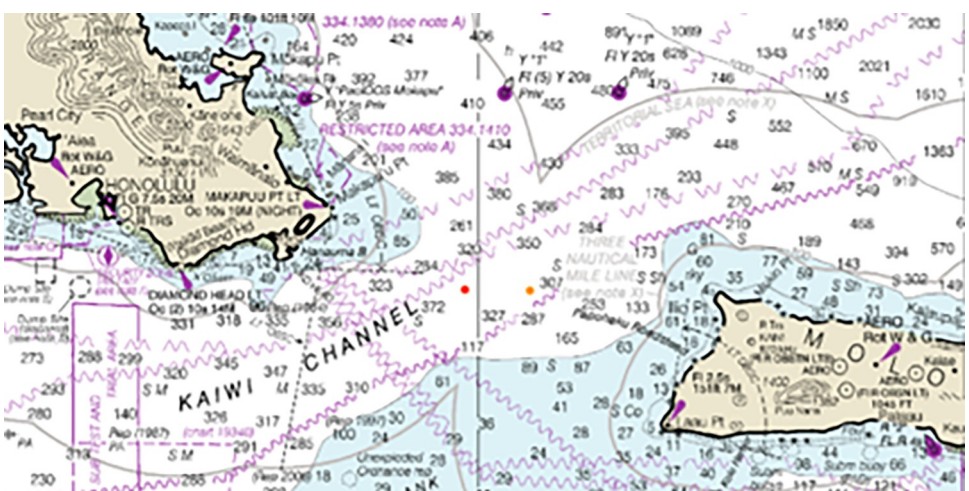

**Fig 2. Map with depth isobaths in fathoms showing the locations of attacks in Kaiwi Channel (Red dot victims 2,3,4; orange dot victim 6).** Provided by National Oceanic and Atmospheric Administration (NOAA) Office of Coast Survey, nauticalcharts.noaa.gov.

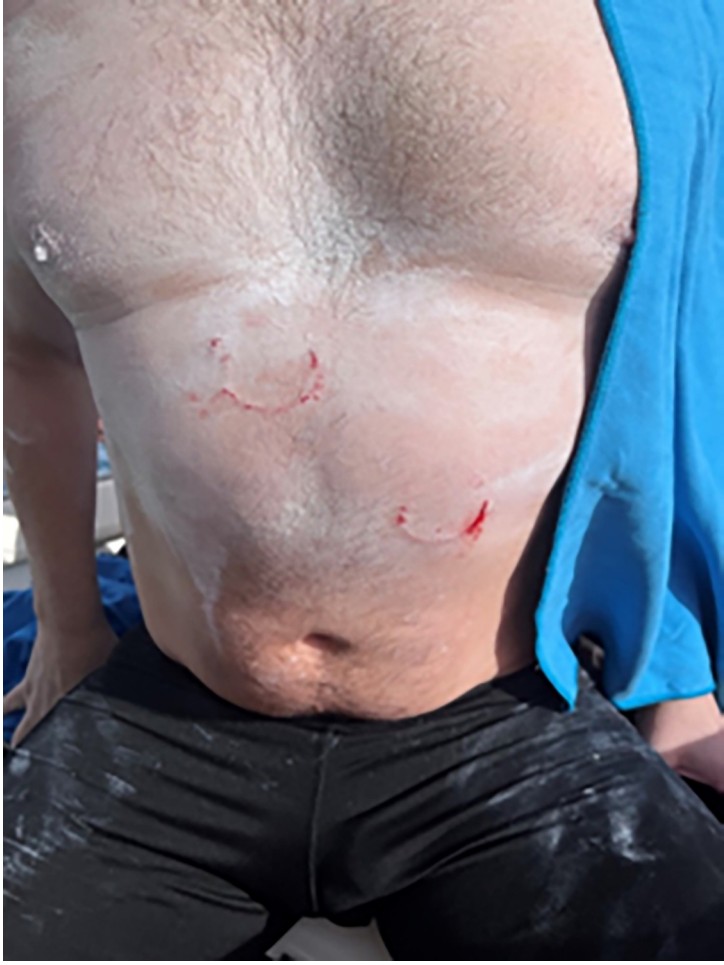

**Fig 3. Photograph of the one successful swimmer bitten twice by *Isistius spp*.** Consent to publication was obtained from the subject.

**Table 3. Variables by sky.**

| | Dark (N = 32) | | Light (N = 101) | | |
|---|---|---|---|---|---|
| | N Eval | N (%) | N Eval | N (%) | P-value |
| Gender | 32 | | 101 | | 0.999 |
| Male | | 21 (66) | | 66 (65) | |
| Female | | 11 (34) | | 35 (35) | |
| Bite | 32 | | 101 | | 0.012 |
| No | | 28 (88) | | 100 (99) | |
| Yes | | 4 (12) | | 1 (1) | |
| Shark deterrent device | 29 | | 89 | | 0.135 |
| No | | 10 (34) | | 46 (52) | |
| Yes | | 19 (66) | | 43 (48) | |
| Artificial illumination (sun down only) | 27 | | 38 | | 0.507 |
| No | | 0 (0) | | 2 (5) | |
| Yes | | 27 (100) | | 36 (95) | |

consumes smaller prey including squid [4–7]. It is possible that this species attacks in groups as Fig 3 and numerous other reports show different diameter bites in live and deceased victims [1, 7–9]. The purple back squid, *Sthenoteuthis oualaniensis* often aggregates at night near boats where the water is illuminated and feeds on other cephalopods and myctophids [22]. *S. oualaniensis* is commonly caught at night in Hawaiʻian waters by shining lights into the water and jigging. Through the process of positive phototaxis the use of artificial lights at night likely

**Table 4. Variables by bite.**

| | No Bite (N = 128) | | Bite (N = 5) | | |
|---|---|---|---|---|---|
| | N Eval | N (%) | N Eval | N (%) | P-value |
| Gender | 128 | | 5 | | 0.163 |
| Male | | 82 (64) | | 5 (100) | |
| Female | | 46 (36) | | 0 (0) | |
| Shark deterrent device | 113 | | 5 | | 0.667 |
| No | | 53 (47) | | 3 (60) | |
| Yes | | 60 (53) | | 2 (40) | |
| Artificial illumination | 108 | | 5 | | 0.320 |
| No | | 31 (29) | | 0 (0) | |
| Yes | | 77 (71) | | 5 (100) | |
| Sun | 128 | | 5 | | 0.068 |
| Sun down | | 70 (55) | | 5 (100) | |
| Sun up | | 58 (45) | | 0 (0) | |
| Sun down only | 70 | | 5 | | 0.157 |
| Dark | | 28 (40) | | 4 (80) | |
| Light | | 42 (60) | | 1 (20) | |
| Year of victim injury | NA | | 6* | | – |
| 2009 | | – | | 1 (17) | |
| 2019 | | – | | 3 (50) | |
| 2022 | | – | | 1 (17) | |
| 2023 | | – | | 1 (17) | |

*includes non-channel swimmer

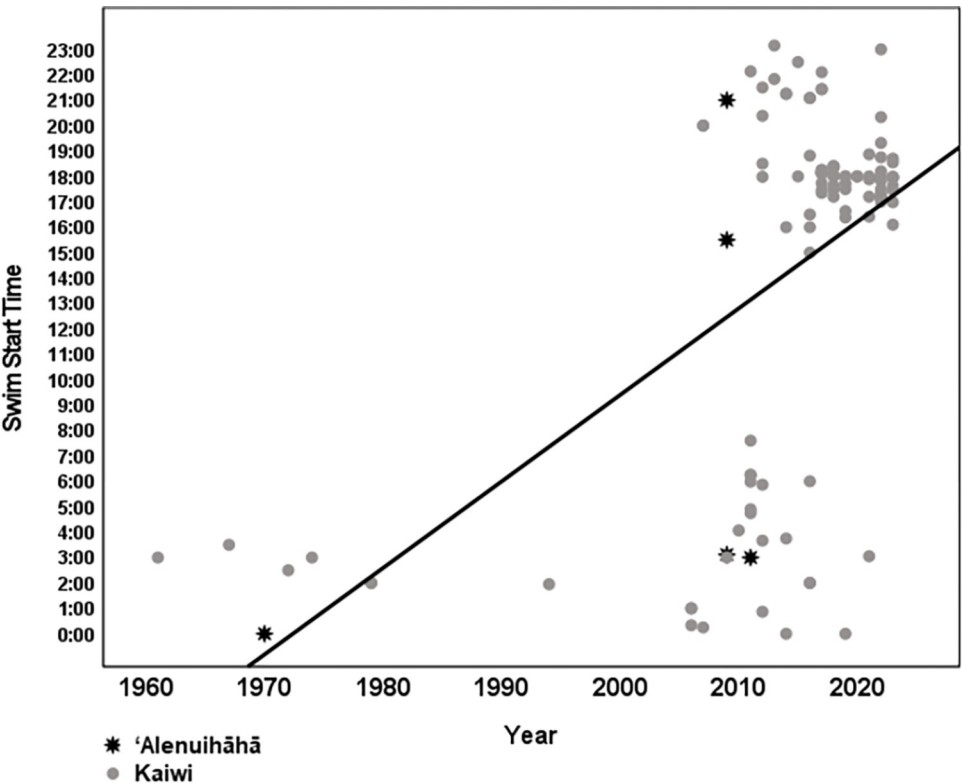

**Fig 4. Swim start time by year for ʻAlenuihāhā and Kaiwi Channels.**

contribute to feeding opportunities for *Isistius spp*. by attracting prey items. Anecdotally, deep-water fisherman that fish in Hawaiʻian waters at night often indicate that the darker the night sky the greater the ability of artificial lights to aggregate fish. In contrast, surface organisms are more scattered during periods of natural illumination and artificial lights do not possess the same aggregative ability. A hypothesis generated from this information is that on the darkest nights, the artificially lit human activity more effectively lures prey items of *Isistius spp*. to the surface thus creating an opportunity for the species to come in contact with live humans swimming at the surface. It is common for swimmers to wear artificial illumination on their suits, goggle straps, and caps and for accompanying kayakers to display artificial illumination for safety purposes and to avoid being lost at sea. It is possible that the addition of artificial illumination may contribute to aggregation of *Isistius spp*. prey items in otherwise dark conditions.

Fig 4 shows that around 2010 it became common for swimmers to start in the late afternoon or early evening. It is highly likely that this change in decision making intended to reduce sun exposure, the likelihood of a nighttime finish, and the impact of strong coastal winds that arise after sunrise while unintentionally increasing the likelihood that swimmers transit the zoogeographical habitat of Isistius spp. in the dark. It is clear from this analysis that swims started occurring more frequently with the sun down after 2010. An analysis of moon phase showed that there is no correlation between moon phase in days and year of swim (Fig 5). While tidal data was not collected it is unlikely to have had a significant impact on this analysis. Hawaiʻi like most of the Pacific has tides with strong and variable inequality meaning there is a difference in height between the two daily high tides and the two daily low tides. In addition, the timing of these tides changes every day. Therefore, if departing on a high or low tide was a

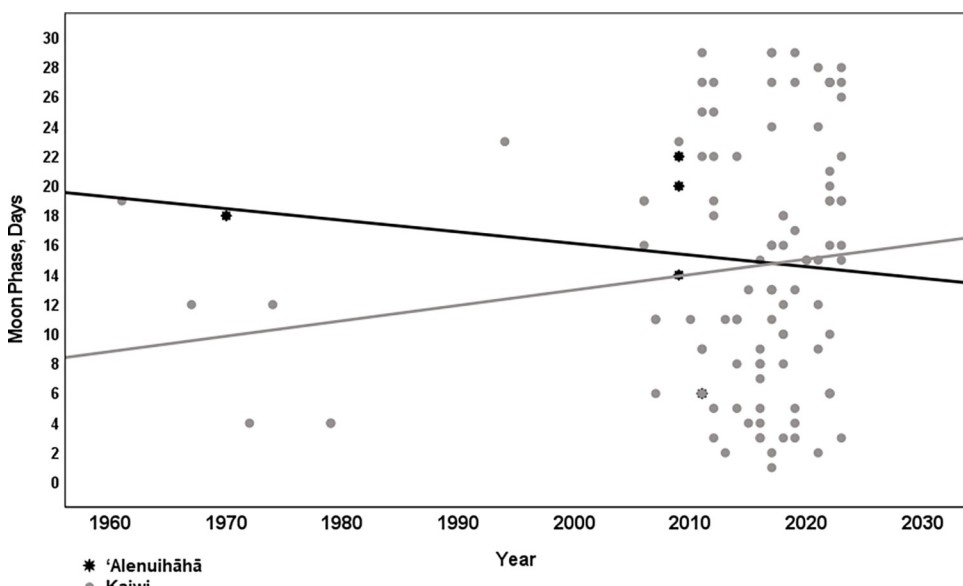

**Fig 5. Moon phase in days by year for ʻAlenuihāhā and Kaiwi Channels.**

primary objective then start times would not likely vary by year as they did and in fact be more evenly distributed across year as well as time of day.

Our data suggests that swimmers crossing the deepest portions of the Kaiwi and ʻAlenuihāhā Channels at night and during periods of no moon are at increased risk of contact with *Isistius spp*. This probability risk would marginally increase if the individual not swimming a channel while attacked in the ʻAlenuihāhā Channel with the moon set and after nautical twilight end and before nautical twilight start was included in the analysis. Indeed, there have never been any documented *Isistius spp*. bites on any other swimmers including the other 691 channel swimmers that successfully completed any of the ʻAuʻau, Pailolo, Kalohi, and Alalakeiki Channels in Hawaiʻi with a maximum surface over water depth of 251 meters [16–18]. Of note, several single, double, and triple cross swims in these shorter channels have included night swimming [16, 23, 24].

Strengths of this study include the definition of the zoogeographical habitat of *Isistius spp*. determined by local sources and current knowledge of species behavior. The definition of this habitat could be significantly improved in the future through electronic tagging of cookiecutter sharks instead of relying on the reported surface over water depth from 6 attacks. Additional strengths include the strict criteria used to determine the "light" versus "dark" groups of swimmers transiting this area. Swimmers were placed in the "dark" group if they transited the habitat in the dark with the moon down for any length of time. This decision was made to reflect exposure to the risk variable. Limitations include a very small sample size in the 'bite group' rendering these data preliminary and missing data from 22 swimmers and/or crew only pertaining to the use of artificial illumination and/or shark deterrent devices. Another limitation includes the methodology used to determine when swimmers entered and exited the zoogeographical habitat of *Isistius spp*. Times were calculated based on the average speed of each swimmer although it is possible that speed varied due to current and tidal factors. It is unlikely that this methodology affected any particular group disproportionately. More accurate data can be made available through the use of global positioning devices to better pinpoint a swimmer's location in the channel at each time point, however use of these devices was not consistent among crews even in recent years. Prospective collection of this data is underway.

## Conclusion

Periods of no natural light when both the moon and sun are set are associated with increased frequency of live human and *Isistius spp*. contact compared to moonlit nights and daytime. This is in stark contrast to prior reports suggesting moonlit nights confer greater risk. These data further support *Isistius spp*. zoogeographical data indicating that the species lives in marine environments with depths of at least 366 meters and are nocturnal hunters. It is possible that the presence of artificial sources of illumination in the darkest conditions may lead to increased *Isistius spp*. and live human interaction although this retrospective cohort study did not support this assumption. In addition to considering sea and weather conditions and sun exposure duration, a strong recommendation supporting human safety can be made to structure channel swimming so that a swimmer transits through the zoogeographical habitat *Isistius spp*. during moonlit nights or daytime.

## Supporting information

**S1 Data.**
(XLSX)

**S1 File.**
(PDF)

## Acknowledgments

The authors thank Carl Kawauchi, Michael Twigg-Smith, Bill Goding, Ivan Shigaki, Mike Spalding, Adherbal de Oliveira, Eric Schall, Isaiah Mojica, Andy Walberer, Steven Munatones, Dr. Harry Huffaker, Dr. Patrick Pedro, Dr. Darryl Takebayashi, and the swimmers of the Hawaiʻian channels for providing valuable information necessary to complete this research.

## Author Contributions

**Conceptualization:** Steven Minaglia.

**Data curation:** Steven Minaglia.

**Formal analysis:** Melodee Liegl.

**Investigation:** Steven Minaglia.

**Methodology:** Steven Minaglia.

**Project administration:** Steven Minaglia.

**Resources:** Steven Minaglia, Melodee Liegl.

**Software:** Melodee Liegl.

**Supervision:** Steven Minaglia.

**Validation:** Melodee Liegl.

**Visualization:** Melodee Liegl.

**Writing – original draft:** Steven Minaglia.

**Writing – review & editing:** Steven Minaglia.

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
