## [Decision Letter · Decision Letter 0]

9 Nov 2023

PONE-D-23-28667Moonless night sky increases Isistius species and live human contactPLOS ONE

Dear Dr. Minaglia,

Thank you for submitting your manuscript to PLOS ONE. After careful consideration, we feel that it has merit but does not fully meet PLOS ONE’s publication criteria as it currently stands. Therefore, we invite you to submit a revised version of the manuscript that addresses the points raised during the review process.

We look forward to receiving your revised manuscript.

Kind regards,

Jianhong Zhou

Staff Editor

PLOS ONE

Journal Requirements:

2. Thank you for including your ethics statement:  "Data for all swimmers was obtained from previously published sources (Citations: 1,2,11,12,14,15) and therefore Institutional Review Board approval was not applicable".   

For studies reporting research involving human participants, PLOS ONE requires authors to confirm that this specific study was reviewed and approved by an institutional review board (ethics committee) before the study began. Please provide the specific name of the ethics committee/IRB that approved your study, or explain why you did not seek approval in this case.

Reviewers' comments:

Reviewer's Responses to Questions

**Comments to the Author**

1. Is the manuscript technically sound, and do the data support the conclusions?

Reviewer #1: Yes

Reviewer #2: Partly

2. Has the statistical analysis been performed appropriately and rigorously? 

Reviewer #1: Yes

Reviewer #2: Yes

3. Have the authors made all data underlying the findings in their manuscript fully available?

Reviewer #1: Yes

Reviewer #2: Yes

4. Is the manuscript presented in an intelligible fashion and written in standard English?

Reviewer #1: Yes

Reviewer #2: Yes

5. Review Comments to the Author

Reviewer #1: Line 27- It can be used Isistius brasiliensis and I. plutodus (Isisitius spp.) and therefore use only Isistius spp to refer to both species along the text.

Lines 196 to 197 – The name Isisitus spp appear 3 times in the same phrase. Maybe it is possible to reduce.

Figure 1. It is possible to see two shark bites. I missed something in the discussion regarding the behavior to attack in groups or repetitive attempts by the same shark. There is some information in the available literature.

Figure 2. It looks like there is a gap between the 1980’s and 2000’s. Why? Is there any reason for that? It is clear that the presented tendency was leaded by the Kaiwi data.

I missed a map with depth isobaths showing the locations of such attacks, or even the location of the channels.

Reviewer #2: Line 53: Denote this is their daytime depth range for species thought to undertake diel vertical migrations.

Line 55: Return to bathyal zone (1,000-4,000 m).

Line 145-146: Can the authors provide any more specifics regarding what type of shark deterrents (e.g., magnetic, electrical, etc.) or make/model were used?

Line 159: Marginally non-significant when p>0.05 (p = 0.068). Moreover, with such a large discrepancy in sample size between the no-bite (n = 128) and bite (n= 5) groups, even a statistically significant result may not be practically meaningful. Limitations in sample size and statistical analysis should be described in the Discussion, as these results are more preliminary than conclusive. 5/133 or 3.75% is a relatively low probability of bite risk and thus challenging to draw meaningful conclusions as to the underlying drivers of these bites. For example, the only statistically significant result was for the swimmers bitten by Isistius spp. in the dark group 4 (12%) versus the light group 1 (1%) with the 95% confidence interval of relative risk spanning 98.5% of the data (1.5-108.9%).

Lines 161-167: It is not entirely clear what the purpose of this analysis is in the context of the manuscript. While the authors report a moderately strong positive correlation between the start time of swim and year of channel swim, it is more apparent that some channel swimmers started their swim in the afternoon/evening (16:00-23:00) after ~2005, though, other channel swimmers started their swim around the same time of day (01:00-06:00) prior to 2005. Start times are more likely correlated with the timing of the sun, moon, and tides and shifts in channel swimming strategies than the year. I recommend either (1) elaborating more on the implications of this analysis, (2) conducting analyses on the timing of swim with the timing of the sun, moon, and tide, and/or (3) removing this analysis.

Lines 170-180: Most of this paragraph seems better suited for the Results section rather than the Discussion since it is reporting numerous descriptive statistics.

Line 185-186: Recommend including the following reference:

Papastamatiou YP, Wetherbee BM, O’Sullivan J, Goodmanlowe GD, Lowe CG. 2010. Foraging ecology of cookiecutter sharks (Isistius brasiliensis) on pelagic fishes in Hawaii, inferred from prey bite wounds. Environmental Biology of Fishes. 88: 361-368.

Line 203: “Goggle” straps instead of “google”?

Lines 206-208: Again, greatest risk of contact is drawn from a very low sample size (n=4/133). Therefore, the sample size limitations of this study should be raised, and I recommend weakening this statement (and Lines 232-24) a bit more.

Lines 208-210: Again, including one more individual would not make the previous statement or data any more statistically robust, but rather very marginally increase the probability risk.

Line 216: I recommend including text on how the definition of the zoogeographical habitat could be significantly improved in the future through electronic tagging of cookie cutter sharks rather than relying on the reported surface over water depth from 6 attacks, as these represent snapshots in time and space.

Lines 239-241: The timing of sea state conditions and sun exposure, as you previously described on Lines 198-201, is also very important to consider, if not the most important consideration given the relatively low Isistius spp. attacks, with regards to human safety. Thus, optimal conditions may not overlap entirely with daylight hours.

6. PLOS authors have the option to publish the peer review history of their article (what does this mean?). If published, this will include your full peer review and any attached files.

Reviewer #1: No

Reviewer #2: No

---

## [Author Response · Author response to Decision Letter 0]

22 Dec 2023

Thank you for the opportunity to revise our manuscript. 

Editor:

#1. 1. When submitting your revision, we need you to address these additional requirements.

Done

#2. Thank you for including your ethics statement: "Data for all swimmers was obtained from previously published sources (Citations: 1,2,11,12,14,15) and therefore Institutional Review Board approval was not applicable". 

For studies reporting research involving human participants, PLOS ONE requires authors to confirm that this specific study was reviewed and approved by an institutional review board (ethics committee) before the study began. Please provide the specific name of the ethics committee/IRB that approved your study, or explain why you did not seek approval in this case.

The following has been inserted, “With the exception of one swimmer, data for all swimmers including victims of attacks was obtained from publically available, previously published sources 1,2,12,13,15,16. Specifically, the date of swim, duration of swim, gender, start and end times, and use of artificial lights and/or electronic shark deterrents were known for successful swimmers and are part of an established database used to certify and celebrate the accomplishment of the swimmers.16 Additionally, data for 4 of the victims was complete and obtained from the case report and case series that described their attacks. 1,2 Written informed consent was obtained from the sole successful yet injured swimmer described in this report (Fig 3., Table 2: Victim 6) whom provided additional data that was not publically available at the time this study was performed. The Institutional Review Board policy of the Queen’s Medical Center does not require approval for use of the information pertaining to the one swimmer.”

NOTE TO EDITORS: You can find evidence of this public display of accomplishment at all 3 of these public websites. Please let us know if you wish for us to include all 3 as references:

http://www.hawaiiswim.org/hawaiianChannel/kaiwiChannel.html

#3. 3. In your Data Availability statement, you have not specified where the minimal data set underlying the results described in your manuscript can be found. PLOS defines a study's minimal data set as the underlying data used to reach the conclusions drawn in the manuscript and any additional data required to replicate the reported study findings in their entirety. All PLOS journals require that the minimal data set be made fully available. For more information about our data policy, please see http://journals.plos.org/plosone/s/data-availability.

The minimal data set used in this manuscript has been attached to the resubmission of this manuscript as “Supporting Information File_Minimal Data Set”.

Reviewer #1: Thank you for your comments.

Reviewer #1: Line 27- It can be used Isistius brasiliensis and I. plutodus (Isisitius spp.) and therefore use only Isistius spp to refer to both species along the text.

Response: We agree. Isistius spp. is used throughout the text after both species are named and collectively identified in line 27. Use of both terms has been deleted from other areas of the text.

Lines 196 to 197 – The name Isisitus spp appear 3 times in the same phrase. Maybe it is possible to reduce

Response: We have revised the prior statement to, “A hypothesis generated from this information is that on the darkest nights, the artificially lit human activity more effectively lures prey items of Isistius spp. to the surface thus creating an opportunity for the species to come in contact with live humans swimming at the surface”.

Figure 1. It is possible to see two shark bites. I missed something in the discussion regarding the behavior to attack in groups or repetitive attempts by the same shark. There is some information in the available literature.

Response: Citations 1,7,8,9 all include data showing different diameter bites indicating involvement by more than one shark in a live victim and several deceased victims. The following has been added to line 220, “It is possible that this species attacks in groups as figure 1 and numerous other reports show different diameter bites in live and deceased victims. “

Figure 2. It looks like there is a gap between the 1980’s and 2000’s. Why? Is there any reason for that? It is clear that the presented tendency was leaded by the Kaiwi data.

Response: We cannot speculate on why there was a gap. It is possible that marathon swimming in Hawaii did not gain popularity until the 2000s.

I missed a map with depth isobaths showing the locations of such attacks, or even the location of the channels.

Response: Maps with depth isobaths showing locations of attacks and channels have been included as Fig 1,2.

Reviewer #2: Thank you for your comments.

 Line 53: Denote this is their daytime depth range for species thought to undertake diel vertical migrations.

Line 55: Return to bathyal zone (1,000-4,000 m).

Response: Cookiecutter sharks are thought to undertake diel vertical migrations with a daytime depth range between 1,000-3,500 meters in primarily warm, coastal waters near islands.4 They typically rise near the surface at night to feed on squid, small fish, and a variety of large pelagic species and then return to the bathyal zone when daylight resumes. 5,6

Line 145-146: Can the authors provide any more specifics regarding what type of shark deterrents (e.g., magnetic, electrical, etc.) or make/model were used?

Response: 123 swimmers and/or crew (92%) were successfully contacted to obtain the additional data pertaining to use of artificial illumination and/or electronic shark deterrent devices. 

Line 159: Marginally non-significant when p>0.05 (p = 0.068). Moreover, with such a large discrepancy in sample size between the no-bite (n = 128) and bite (n= 5) groups, even a statistically significant result may not be practically meaningful. Limitations in sample size and statistical analysis should be described in the Discussion, as these results are more preliminary than conclusive. 5/133 or 3.75% is a relatively low probability of bite risk and thus challenging to draw meaningful conclusions as to the underlying drivers of these bites. For example, the only statistically significant result was for the swimmers bitten by Isistius spp. in the dark group 4 (12%) versus the light group 1 (1%) with the 95% confidence interval of relative risk spanning 98.5% of the data (1.5-108.9%).

Response: Limitations in sample size and statistical analysis are further described in the discussion. The following statement has been added to the discussion, “Limitations include a very small sample size in the ‘bite group’ rendering these data preliminary and missing data from 22 swimmers and/or crew only pertaining to the use of artificial illumination and/or shark deterrent devices.”

Lines 161-167: It is not entirely clear what the purpose of this analysis is in the context of the manuscript. While the authors report a moderately strong positive correlation between the start time of swim and year of channel swim, it is more apparent that some channel swimmers started their swim in the afternoon/evening (16:00-23:00) after ~2005, though, other channel swimmers started their swim around the same time of day (01:00-06:00) prior to 2005. Start times are more likely correlated with the timing of the sun, moon, and tides and shifts in channel swimming strategies than the year. I recommend either (1) elaborating more on the implications of this analysis, (2) conducting analyses on the timing of swim with the timing of the sun, moon, and tide, and/or (3) removing this analysis.

Response: 

The following has been added to the results, “There was no significant correlation between moon phase in days and year of swim for ’Alenuihāhā channel (p=0.723) and Kaiwi channel (p=0.140) and both combined (p=0.225)(Figure 3).”

The following has been added to the discussion section along with Figure 3 to Supplemental information, “Figure 2 shows that around 2010 it became common for swimmers to start in the late afternoon or early evening. It is highly likely that this change in decision making intended to reduce sun exposure, the likelihood of a nighttime finish, and the impact of strong coastal winds that arise after sunrise while unintentionally increasing the likelihood that swimmers transit the zoogeographical habitat of Isistius spp. in the dark. It is clear from this analysis that swims started occurring more frequently with the sun down after 2010. An analysis of moon phase showed that there is no correlation between moon phase in days and year of swim (Figure 3). While tidal data was not collected it is unlikely to have had a significant impact on this analysis. Hawaiʻi like most of the Pacific has tides with strong and variable inequality meaning there is a difference in height between the two daily high tides and the two daily low tides. In addition, the timing of these tides changes every day. Therefore, if departing on a high or low tide was a primary objective then start times would not likely vary by year as they did and in fact be more evenly distributed across year as well as time of day.

Lines 170-180: Most of this paragraph seems better suited for the Results section rather than the Discussion since it is reporting numerous descriptive statistics.

Response: The following two paragraphs were edited based on these recommendations, “The two channels, Kaulakahi and Kealaikahiki are both 17 miles and on average took swimmers 11 hours 17 minutes (range 9:25-14:28) and 11 hours 22 minutes (range 10:24-11:53) to cross, respectively. The ʻAlenuihāhā and Kaiwi Channels are 30 and 26 miles in length and have average crossing times of 17 hours 26 minutes (range 14:51-20:08) and 16 hours 40 minutes (12:02-27:33), respectively. Swim start time and year of channel swim was significant when looking at the ’Alenuihāhā and Kaiwi channels only. During the earlier years, the start time was just after midnight versus later years (post 2010) starting closer to 6 PM. Pearson correlation was 0.566, p <0.001 (Figure 2). In contrast, all swim start times were between 6:00 AM and 7:33 AM for the Kealaikahiki Channel and between 4:21 AM and 6:27 AM for the Kaulakahi Channel. This effectively placed all swimmers of these two channels in the zoogeographical habitat of Isistius spp. during daytime.

1. Discussion

The width of each channel determines the average time it takes for a channel swimmer to cross. The Kaiʻeiʻewaho Channel is 72 miles and has only been crossed once by a relay team of 6 swimmers with a total time of 47 hours 55 minutes. This channel is the deepest and given its length it is unlikely to be attempted by a serious solo swimmer. In 2008, the inclusion of the Kaiwi Channel to Oceans 7, a list of 7 long-distance open-water swims through some of the worlds’ most dangerous sea channels has led to a significant increase in swims across this particular channel. It is highly unlikely for any swimmer to complete either of the ’Alenuihāhā and Kaiwi channels exclusively during daylight hours given average crossing times.”

Line 185-186: Recommend including the following reference:

Papastamatiou YP, Wetherbee BM, O’Sullivan J, Goodmanlowe GD, Lowe CG. 2010. Foraging ecology of cookiecutter sharks (Isistius brasiliensis) on pelagic fishes in Hawaii, inferred from prey bite wounds. Environmental Biology of Fishes. 88: 361-368.

Response: Included as reference #7 given earlier mention of a similar statement in the text- thank you for this citation.

Line 203 changed to goggle straps.

Lines 206-208: Again, greatest risk of contact is drawn from a very low sample size (n=4/133). Therefore, the sample size limitations of this study should be raised, and I recommend weakening this statement (and Lines 232-24) a bit more.

Response: We changed the statement to, “Our data suggests that swimmers crossing the deepest portions of the Kaiwi and ʻAlenuihāhā Channels at night and during periods of no moon are at increased risk of contact with Isistius spp.”

Lines 208-210: Again, including one more individual would not make the previous statement or data any more statistically robust, but rather very marginally increase the probability risk.

Response: This statement was changed to, “This probability risk would marginally increase if the individual not swimming a channel while attacked in the ʻAlenuihāhā Channel with the moon set and after nautical twilight end and before nautical twilight start was included in the analysis.”

Line 216: I recommend including text on how the definition of the zoogeographical habitat could be significantly improved in the future through electronic tagging of cookie cutter sharks rather than relying on the reported surface over water depth from 6 attacks, as these represent snapshots in time and space.

Response: “Strengths of this study include the definition of the zoogeographical habitat of Isistius spp. determined by local sources and current knowledge of species behavior. The definition of this habitat could be significantly improved in the future through electronic tagging of cookiecutter sharks instead of relying on the reported surface over water depth from 6 attacks.”

Line 232-234

Response: We changed the statement to, “Periods of no natural light when both the moon and sun are set are associated with increased frequency of live human and Isistius spp. contact compared to moonlit nights and daytime.”

Lines 239-241: The timing of sea state conditions and sun exposure, as you previously described on Lines 198-201, is also very important to consider, if not the most important consideration given the relatively low Isistius spp. attacks, with regards to human safety. Thus, optimal conditions may not overlap entirely with daylight hours.

Response: “In addition to considering sea and weather conditions and sun exposure duration, a strong recommendation supporting human safety can be made to structure channel swimming so that a swimmer transits through the zoogeographical habitat Isistius spp. during moonlit nights or daytime.”

---

## [Editor Report · Decision Letter 1]

21 Jan 2024

Moonless night sky increases Isistius species (cookiecutter shark) and live human contact

PONE-D-23-28667R1

Dear Dr. Minaglia,

We’re pleased to inform you that your manuscript has been judged scientifically suitable for publication and will be formally accepted for publication once it meets all outstanding technical requirements.

Kind regards,

Daniel M Coffey, Ph.D.

Guest Editor

PLOS ONE

Additional Editor Comments (optional):

Dear Dr. Minaglia,

Thank you for ensuring that your manuscript meets PLOS ONE's style requirements, including the Policy on Clinical Case Reports and Case Series from The Queen’s Medical Center Office of Research and Development, and addressing the reviewer's comments and recommended revisions. To maintain transparency and uphold the integrity of the scientific process, I would like to acknowledge that before being invited to serve as the Guest Academic Editor for this submission, I had previously served as a reviewer for your original submission. I am pleased to inform you that your manuscript has been accepted for publication in PLOS ONE. Congratulations!
---

## [Editor Report · Acceptance letter]

26 Jan 2024

PONE-D-23-28667R1 

PLOS ONE

Dear Dr. Minaglia, 

I'm pleased to inform you that your manuscript has been deemed suitable for publication in PLOS ONE. Congratulations! Your manuscript is now being handed over to our production team.

Kind regards, 

on behalf of

Dr. Daniel M Coffey 

Guest Editor

PLOS ONE